# The Use of Specific Non-*Saccharomyces* Yeasts as Sustainable Biocontrol Solutions Against *Botrytis cinerea* on Apples and Strawberries

**DOI:** 10.3390/jof11010026

**Published:** 2025-01-02

**Authors:** Zukisani Gomomo, Morris Fanadzo, Maxwell Mewa-Ngongang, Boredi Silas Chidi, Justin Wallace Hoff, Marieta van der Rijst, Lucky Mokwena, Mathabatha Evodia Setati, Heinrich Wilbur du Plessis

**Affiliations:** 1Post-Harvest and Agro-Processing Technologies Division, ARC Infruitec-Nietvoorbij, Stellenbosch 7599, South Africa; mewam@arc.agric.za (M.M.-N.); hoffjw@arc.agric.za (J.W.H.); 2Department of Agriculture, Cape Peninsula University of Technology, Wellington 7654, South Africa; fanadzom@cput.ac.za; 3Department of Biotechnology and Consumer Sciences, Cape Peninsula University of Technology, Cape Town 8000, South Africa; chidib@cput.ac.za; 4ARC Biometry, ARC Infruitec-Nietvoorbij, Stellenbosch 7599, South Africa; vanderrijstm@arc.agric.za; 5Central Analytical Facilities, Stellenbosch University, Stellenbosch 7602, South Africa; mokwenal@sun.ac.za; 6Department of Viticulture and Oenology, South African Grape and Wine Research Institute, Stellenbosch University, Stellenbosch 7602, South Africa; setati@sun.ac.za

**Keywords:** mould spoilage, synthetic chemicals, pre- and post-harvest control, growth inhibition, hydrolytic enzymes

## Abstract

Apples and strawberries hold significant commercial and nutritional value but face pre- and post-harvest spoilage due to infections by *Botrytis cinerea.* While spoilage is conventionally managed using synthetic chemicals, there is a growing interest in utilising yeasts as biological control agents. This study aimed to assess the antifungal potential of non-*Saccharomyces* yeasts *Suhomyces pyralidae, Meyerozyma guilliermondii, Pichia kluyveri*, *Zygoascus hellenicus*, and *Aureobasidium melanogenum* against three *B. cinerea* strains (B05.10, IWBT-FF1, and PPRI 30807) on agar plates and in post-harvest trials on apples and strawberries. *Aureobasidium melanogenum* exhibited a broad range of extracellular enzyme production and inhibition rates of 55%, 52%, and 40% against the strains. In volatile organic compound (VOC) assays, *P. kluyveri* and *S. pyralidae* achieved 79% and 56% inhibition, respectively, with VOCs like isobutanol, isoamyl alcohol, 2-phenylethanol, isoamyl acetate, and 2-phenethyl acetate identified. In post-harvest trials, *S. pyralidae* was most effective on apples, with inhibition rates up to of 64%. The commercial fungicide Captan and *S. pyralidae* and *P. kluyveri* achieved 100% inhibition against the *B. cinerea* strains B05.10 and IWBT-FF1 on strawberries. These findings highlight the potential of the selected yeast species as biological control agents against *B. cinerea*, warranting further research into their application in commercial fruit protection.

## 1. Introduction

Apples (*Malus domestica* (Suckow) Borkh.) and strawberries (*Fragaria × ananassa* (Duchesne ex Weston) Duchesne ex Rozier) are valuable for human health, serving as primary sources of essential nutrients, including vitamins and minerals, that support a healthy lifestyle [1,2,3,4]. Their appealing sensory and nutritional characteristics make them widely consumed and processed into various products, such as cooked slices, juices, and jellies, contributing significantly to the global fresh produce market [3,5,6]. Despite their importance as major fruit crops worldwide, producers continue to encounter numerous challenges in production, storage, and market distribution [7,8].

Commercially grown fruits are destined for both local and export markets after harvest. However, pre- and post-harvest mould decay leads to significant economic losses [3,8]. Mould infections during these stages are often attributed to elevated moisture levels, excessive nutrients, low pH, and reduced fruit decay resistance as maturity progresses [9]. Among the most severe diseases affecting strawberries and apples is grey mould, caused by *Botrytis cinerea*, which significantly impacts yield and quality by depleting nutrients, shortening shelf life, and causing substantial financial losses [3,4,10,11]. Grey mould infection typically initiates during flowering, remaining latent until fruit maturation, at which point the pathogen proliferates extensively [12,13].

The control of *B. cinerea* Pers. (1794) presents a significant challenge due to the pathogen’s high genetic plasticity, with chemical control using synthetic fungicides being the most widely used strategy [6,14,15]. While synthetic fungicides effectively reduce the pre- and post-harvest fruit losses, their use has led to increased fungicide residues on produce, the emergence of fungicide-resistant mould strains, and has raised concerns regarding human health and the environmental impact [3,4,5]. Consequently, it is crucial to develop safe and effective alternative strategies for managing grey mould diseases in fruit crops [3,16].

Antagonistic fungi have proven effective against *B. cinerea*, which is susceptible to suppression by microorganisms such as non-*Saccharomyces* yeasts that produce antifungal compounds [3,6,17]. Compared to chemical control, the use of antagonist microorganisms offers several benefits, including the absence of toxic residues, environmental safety, ease of application, and cost-effectiveness [15,18]. Yeasts exhibit valuable antifungal properties, including the secretion of killer toxins such as mycocins, the production of cell wall-degrading enzymes (chitinase, *β*-1,3-glucanase, protease, laminarinases, and peroxidases), the synthesis of volatile organic compounds (VOCs), rapid colony formation, growth within surface wounds, competition for nutrients and space, and the induction of host resistance [3,13,14]. Killer yeasts, including the *Meyerozyma guilliermondii, Suhomyces pyralidae* (formerly, *Candida pyralidae*), *Pichia kluyveri*, and *Hanseniaspora* species, have shown antimicrobial activity against a range of fruit-spoilage fungi [17,19,20,21]. Previous research by Gomomo et al. [17] evaluated non-*Saccharomyces* yeasts for their ability to inhibit mycelial growth of a strain of *B. cinerea* in vitro and on apples, with results indicating species- and strain-dependent inhibitory effects. Building on this foundation, the present study sought to screen non-*Saccharomyces* yeasts for extracellular enzyme activity and to assess the inhibition of the mycelial growth and spore germination of selected yeasts against three distinct *B. cinerea* strains in vitro and in vivo on apples and strawberries. The yeasts that showed the most potential as biocontrol agents according to Gomomo et al. [17] were used in this study to confirm their biocontrol activity against *B. cinerea.*

## 2. Materials and Methods

### 2.1. Culturing Conditions and Inoculum Preparation

Twenty-three yeast isolates (Table 1) were sourced from the biobank of ARC Infruitec-Nietvoorbij (Fruit, Vine and Wine Institute of the Agricultural Research Council, Stellenbosch, South Africa). The selected yeast strains performed the best as potential biocontrol agents according to previous research findings from Gomomo et al. [17] and were selected for enzyme activity screening. The yeasts were initially cultured on yeast malt agar (YMA) media composed of 1% glucose, 0.3% malt extract, 0.3% yeast extract, 0.5% peptone, and 2% bacteriological agar and incubated at 28 °C for 48 h. For inoculum preparation, a loopful of each pure yeast colony was transferred into test tubes containing 5 mL of sterilised yeast malt broth (YMB) (Sigma-Aldrich, Johannesburg, South Africa) and incubated at 28 °C for another 48 h. Yeast cell counts were then performed using a haemocytometer under a microscope at 400× magnification to standardise the yeast inoculum concentration to 1 × 10^8^ cells/mL.

*Botrytis cinerea* strains B05.10 and IWBT-FF1 were sourced from the South African Grape and Wine Research Institute (Stellenbosch University, South Africa), and isolate PPRI 30807 was acquired from the ARC Plant Health and Protection biobank (ARC Plant Health and Protection Institute, Pretoria, South Africa). The mould cultures were grown on potato dextrose agar (PDA, Merck, Johannesburg, South Africa) at 25 °C for 7 to 14 days. To prepare inoculum, a 5 mm mycelial disc was excised from a 5-day-old culture plate for each strain. Spores were harvested by gently scraping the plate surface with a sterile loop and rinsing with sterile distilled water to obtain a 50 mL spore suspension, which was collected in a sterile 250 mL Schott bottle. The spore concentration was adjusted to 1 × 10^5^ spores/mL using a haemocytometer and microscope at 400× magnification.

### 2.2. Extracellular Lytic Enzyme Activity

The yeast isolates were evaluated for their ability to produce lytic enzymes, including proteases, chitinases, glucanases, cellulase, starch-degrading amylases, pectinase, and lipases. A 10 µL suspension of each yeast culture (±1 × 10^8^ cells/mL) was spotted onto agar plates containing specific substrates for each enzyme assay (Figure 1). The plates were incubated at 28 °C for 4–7 days, after which enzymatic activity was assessed. Each treatment was conducted in triplicate. Enzyme activity was indicated by clear halos surrounding the yeast colonies (Figure 1), and was recorded as either (−) for no activity or (+) for activity.

#### 2.2.1. Protease Activity

Protease activity was assessed using a modified protocol based on Liu et al. [22]. Assays were conducted on skim milk agar plates containing 10% skim milk powder and 2% bacteriological agar. Enzymatic activity was indicated by a clear halo around the inoculated area.

#### 2.2.2. Chitinase Activity

Chitinase activity was determined following an adapted method from Verma and Garg [23]. Chitin agar plates, prepared with 0.1% finely ground chitin derived from shrimp as the sole carbon source and 2% bacteriological agar, were used for the assay. After incubation, Gram’s iodine was applied to the plates for 30 min. Chitinase activity was identified by the appearance of clear halos around the colonies.

#### 2.2.3. Glucanase Activity

*β*-1,3-Glucanase activity was evaluated using a laminarin medium consisting of 0.5% laminarin, 0.67% yeast nitrogen base, and 2% bacteriological agar) (Sigma-Aldrich, Johannesburg, South Africa) as following the method described by Strauss et al. [24]. After incubation, the plates were stained with 0.06% Congo red for 60 min at room temperature, and excess stain was decanted. The plates were subsequently treated with 1 mol/L NaCl for 15 min. The enzymatic hydrolysis of glucan was indicated by a yellow-orange halo surrounding the colonies.

#### 2.2.4. Glucosidase Activity

*β*-Glucosidase activity was determined on a selective medium containing 0.67% yeast nitrogen base (YNB, Difco), 0.5% arbutin, and 2% bacteriological agar, as outlined by Strauss et al. [24]. The pH of the medium was adjusted to 5 before autoclaving. Additionally, 10 mL of a 1% ammonium ferric citrate solution (filter-sterilised) was added to the medium before plating. Colonies exhibiting *β*-glucosidase activity were distinguished by a brown discolouration of the medium.

#### 2.2.5. Cellulase Activity

Cellulase activity was evaluated on a medium containing 0.2% carboxymethyl cellulose (CMC), 1% yeast nitrogen base (1%), and 2% bacteriological agar. Following incubation, the plates were flooded with Gram’s iodine for 30 min. Cellulase activity was indicated by clear halos around the colonies.

#### 2.2.6. Starch Degrading Activity

Yeasts were screened for starch-degrading activity on a medium comprising 0.67% YNB, 0.2% soluble starch, and 0.2% bacteriological agar at pH 6 following the protocol by Buzzini and Martini [25]. After incubation, the plates were treated with an iodine solution, and starch hydrolysis was indicated by a pale-yellow zone surrounding the colonies.

#### 2.2.7. Pectinase Activity

Pectinase activity was assessed following an adapted method from McKay [26] using a pectinase agar medium containing 1.25% pectin (Sigma), 0.68% potassium phosphate (pH 3.5), 0.67% YNB, 1% glucose, and 2% bacteriological agar. Plates were stained with 0.1% ruthenium red, and colonies producing a purple halo were identified as positive for pectinase activity.

#### 2.2.8. Lipase Activity

Lipase activity was tested on a tributyrin agar medium containing 0.5% peptone, 0.3% yeast extract, 1% tributyrin, and 2% bacteriological agar adjusted to pH 6, following the method of Buzzini and Martini [25]. A clear halo surrounding the colony in the opaque medium signified lipase activity.

### 2.3. Dual-Culture Assay

Dual-culture assays were used to assess the inhibitory effects of yeasts on mycelial growth, following the protocol by Chen et al. [27]. Four yeast strains, which displayed multiple lytic enzyme activities in initial screenings and showed the best biocontrol activity according to Gomomo et al. [17], were selected for further evaluation (Table 2). Additionally, yeast strain *P. kluyveri* (Y64), previously studied by Mewa-Ngongang et al. [20,28] and Gomomo et al. [17], was included as the reference strain. A 5 mm mycelial disc was positioned at the edge of the YMA plate, and 20 μL of the yeast suspension (1 × 10^8^ cells/mL) was spotted 40 mm away from the mycelial disc (Figure 2). Incubation was conducted at 25 °C for 5–9 days.

Negative control plates contained only the 5 mm diameter mycelial disc of the target mould, while positive control plates included 0.5 g/L of the commercial fungicide Captan (N-trichloromethylthio-4-cyclohexene-1,2-dicarboximide). All treatments were performed in triplicate. The percentage inhibition of mycelial growth (MGI) was calculated using the following formula:MGI = [(D_0_ − D_t_)/D_0_] × 100(1)
with D_0_ representing the average horizontal growth of the mould colony in the negative control and D_t_ representing the average horizontal growth of the fungal colony on the yeast-treated plates (Figure 2), as described by Núñez et al. [29].

### 2.4. Mould Spore Germination Assay

A radial inhibition assay was conducted using the agar plate method as described by Núñez et al. [29]. Yeast cell suspensions (1 × 10^8^ cells/mL) were prepared from yeast culture broths, and 100 µL of each suspension was spread evenly on YMA plates with a Drigalski spatula and allowed to dry. Subsequently, 15 µL of a *B. cinerea* spore suspension (1 × 10^5^ spores/mL) was spotted at the centre of each plate (Figure 3), with each treatment conducted in triplicate. Negative control plates contained only the 15 µL spore suspension at the centre of the YMA. The plates were incubated at 25 °C for 5–9 days. The mould radial inhibition (MRI) was calculated as follows:MRI = [(D_0_ − D_t_)/D_0_] × 100(2)
with D_0_ representing the average diameter of the mould growth on the negative control plates and D_t_ representing the diameter of the mould growth on the yeast-treated plates [29].

### 2.5. Volatile Organic Compound (VOC) Production Assay

The VOC production by selected yeasts was assessed using the mouth-to-mouth assay described by Medina-Córdova et al. [30]. In this assay, two YMA plates were sealed face to face using laboratory film. The bottom plate was spread with 100 µL of the yeast suspension (1 × 10^8^ cell/mL), while a 5 mm mould mycelial disc was placed at the centre of the top plate. For the negative control, only the mycelial disc was placed in the centre of the top plate, with no yeast applied to the bottom plate. For the positive control, 0.5 g/L of the commercial fungicide Captan was spread on one YMA plate, with the mycelial disc placed on the other. The plates were incubated at 25 °C for 7 days, and each treatment was conducted in triplicate. VOC inhibition activity (VOCIA) was calculated using the following mathematical expression [29]:VOCIA = [(D_0_ − D_t_)/D_0_] × 100(3)
with D_0_ representing the average diameter of the mould colony on the negative control plates and D_t_ representing the diameter of the mould colony on the treated plates, as shown in Figure 4.

### 2.6. Extraction of Volatile Organic Compounds and Gas Chromatographic Analyses

#### 2.6.1. Sample Preparation and Analyses

The volatile organic compounds (VOCs) produced by *P. kluyveri* and *S. pyralidae* were analysed using headspace solid-phase microextraction coupled with gas chromatography–mass spectrometry (HS-SPME–GC–MS), following a modified method based on Maluleke et al. [31]. Two sterile YMA layers were prepared by pouring 2 mL of agar on opposite sides of each vial. A spore suspension of *B. cinerea* PPRI 30807 (1 × 10^5^ spores/mL) was prepared and 10 µL was spread on one side of the vial using an inoculation loop (LP ITAKIAN SPA, Milan, Italy). On the opposite side, 10 μL of the yeast suspension (1 × 10^8^ cells/mL) was spread. Vials were incubated at 25 °C for 5 days, with separate control vials inoculated solely with *B. cinerea* spore suspension or the yeast cell suspension. Each treatment, including controls, was performed in triplicate.

For GC–MS analysis, 50 μL of a 10 ppm Anisole d8 solution was added to the centre of each vial as an internal standard. The vials were then incubated in an autosampler at 70 °C for 10 min, after which a 50/30 μm divinylbenzene/carboxen/polydimethylsiloxane (DVB/CAR/PDMS) SPME fibre (Supelco, Bellafonte, PA, USA) was exposed to the headspace of each vial for 30 min at the same temperature. Following equilibration, the fibre was inserted into the GC injector at 250 °C, where compounds were desorbed over 10 min.

#### 2.6.2. Chromatographic Conditions

Analyses were conducted using an Agilent Gas Chromatography, model 6890 N (Agilent, Palo Alto, CA, USA), coupled to an Agilent mass spectrometer detector, model 5975B Inert XL EI/CI (Agilent, Palo Alto, CA, USA), equipped with a CTC Analytics PAL autosampler. Chromatographic separation was achieved on a polar ZBWax capillary column (30 m length, 0.25 mm internal diameter, 0.25 μm film thickness). The oven temperature programme began at 40 °C, held for 17 min, followed by an increase to 240 °C at a rate of 8 °C/min, with a final hold of 5 min. Helium served as the carrier gas at a constant flow rate of 1.0 mL/min. The injector operated in a splitless mode at 250 °C throughout the analysis, with a purge flow of 50 mL/min activated after 2 min and a gas saver flow at 50 mL/min for an additional 5 min. The MS detector operated in a full scan mode, with the ion source and quadrupole temperatures set at 230 °C and 150 °C, respectively, and the transfer line at 280 °C. Compounds were identified by their retention times and mass spectra, referencing the NIST05 spectral library.

### 2.7. Post-Harvest Fruit Bioassays

Post-harvest biocontrol efficacy assays were conducted on “Golden Delicious” apples and “Earliglow” strawberries across 16 treatments (Table 3). Each treatment included five replicates, with each replicate comprising a rectangular fruit-packaging box containing five apples or a punnet with five strawberries. Fruit surfaces were sprayed with 70% ethanol to eliminate surface microorganisms and allowed to dry completely before wound infliction. A sterile cork borer was used to uniformly wound the fruits (approximately 5 mm diameter and 3 mm deep).

After 15 min, 15 μL of sterile purified water was applied to the wound in the control treatment, while the other treatments received 15 μL of the *B. cinerea* spore suspension (1 × 10^5^ spores/mL), followed by a 30 min drying period. Subsequently, 15 μL of a yeast inoculum (1 × 10^8^ cells/mL) or 15 μL of the commercial fungicide Captan (0.5 g/L) was introduced into the wound. The negative control was treated with the three *B. cinerea* spore suspension only without additional yeast or fungicide. The treated fruits were incubated at ±20 °C for 4–6 days at a relative humidity of 80%. The inhibition of mould growth was characterised by an absence of visible mould development. Lesion diameters were measured, and percentage growth inhibition was calculated using previously described formulas.

### 2.8. Statistical Analyses

Percentage inhibition data from each assay were analysed using a one-way analysis of variance (ANOVA) using the GLM procedure of SAS software (version 9.4, SAS Institute Inc, Cary, NC, USA). The normality of standardised residuals was confirmed with the Shapiro–Wilk test. Fisher’s least significant difference (LSD) values were calculated at a 5% significance level (*p* = 0.05) to allow for the comparison of treatment means. A probability level of 5% was deemed significant for all statistical tests.

## 3. Results and Discussion

### 3.1. Extracellular Lytic Enzymes Activity

Among the 23 yeast strains examined, all displayed lipase activity, with additional enzyme activities varying across strains (Table 1). Notably, *A. melanogenum*, produced all the enzymes tested, while *S. pyralidae, Z. hellenicus*, and *M. guilliermondii* Y88 demonstrated starch-degrading enzymes, protease, glucanase, and cellulase activities. *Rhodotorula dairenensis* and *M. guilliermondii* Y65 also produced proteases and glucanases alongside lipases, whereas the remaining yeast strains exhibited activity for one additional enzyme or none.

Previous studies have documented the enzyme-producing capabilities of *Aureobasidium* species. Zajc et al. [10], Parafati et al. [32], and Di Francesco et al. [33] reported glucanase, pectinase, and protease activities for *A. melanogenum, A. pullulans,* and *A. subglaciale*, supporting the role of *Aureobasidium* spp. in producing lytic enzymes. Zajc et al. [10] and Moura et al. [34] further observed chitinase and glucanase activity in *A. melanogenum*, aligning with the current findings. The protease activity in *S. pyralidae* corroborates findings by Kantarcioǧlu and Yücel [12], Oksuz et al. [35], and Mehlomakulu et al. [36].

De Souza Ramos et al. [37] and Yang et al. [38] also reported protease and glucanase activities in *Suhomyces* spp., supporting this study’s observations. Additionally, Ruas et al. [39], Agirman and Erten [40], and Lorrine et al. [41] found *M. guilliermondii* capable of extracellular protease production, although this was strain dependent, consistent with the results here. Maluleke et al. [31] reported that chitinase and glucanase activities were common in yeasts with antagonistic activity against *B. cinerea*, further supporting the findings.

### 3.2. Dual-Culture Assay

Yeasts are known to inhibit various moulds through the production of diffusible metabolites. The five yeasts tested displayed varying levels of growth inhibition against the three *B. cinerea* strains, indicating that inhibition effectiveness varies depending on both yeast species and fungal strain (Figure 5).

*Aureobasidium melanogenum* (Y6) was the most effective, demonstrating 55%, 52%, and 40% inhibition against *B. cinerea* B05.10, IWBT-FF1, and PPRI 30807, respectively, whereas the commercial fungicide Captan achieved 57%, 41%, and 34% inhibition (Figure 5A–C). *Suhomyces pyralidae* (Y63) ranked second, inhibiting *B. cinerea* B05.10, IWBT-FF1, and PPRI 30807 by 56%, 38%, and 35%, respectively. *Meyerozyma guilliermondii* (Y88) inhibited *B. cinerea* strains B05.10, IWBT-FF1, and PPRI 30807 by 53%, 43%, and 15%, respectively. The reference strain *P. kluyveri* (Y64) showed limited inhibition with an average activity of 24% against the three *B. cinerea* strains.

The results of this study align with those of Di Francesco et al. [33], who also reported *A. melanogenum* inhibiting *B. cinerea*, although the yeast strain exhibited lower efficacy in their study compared to this one. Similarly, Gomomo et al. [17] reported that *A. melanogenum* inhibited a different strain of *B. cinerea* by 55% in vitro, supporting the findings of this study. Previous work by Mewa-Ngongang et al. [20] reported 100% inhibition of *S. pyralidae* on *B. cinerea* spore germination, while Gomomo et al. [17] found 62% inhibition under in vitro conditions, suggesting that *S. pyralidae* is more effective at preventing spore germination than controlling established mould growth. Wang et al. [42] and Cheng et al. [43] also reported antifungal activity of *M. guilliermondii* against *B. cinerea*. The reference strain *P. kluyveri* (Y64) showed low inhibition activity, which agrees with observations by Gomomo et al. [17] who noted its relatively weak antagonistic effect. The inhibitory effect of the highest performing yeasts, *A. melanogenum, S. pyralidae*, and *M. guilliermondii*, may be attributed to their production of cell wall-degrading enzymes (Table 1), and their ability to compete with *B. cinerea* for nutrients and space.

### 3.3. Mould Spore Germination Assay

A radial inhibition assay was used to assess the effect of *S. pyralidae, P. kluyveri, A. melanogenum, M. guilliermondii*, and *Z. hellenicus* on the spore germination of three *B. cinerea* strains (Figure 6). Notably, the inhibitory levels observed were higher than those in the dual-culture assay (Figure 5). Previous research by Mewa-Ngongang et al. [20] has shown that these *non-Saccharomyces* yeasts can inhibit mould growth through various mechanisms, such as rapid colonisation of surfaces and outcompeting spoilage moulds, thereby limiting mould proliferation. Both *M. guilliermondii* (Y88) and *P. kluyveri* (Y64) were highly effective, achieving 100% inhibition against all three *B. cinerea* strains (Figure 6A–C). The increased inhibition by *P. kluyveri*, compared to its mycelial growth inhibition in the dual-culture assay (Figure 5), underscores the stronger antagonistic effect of these yeasts on spore germination and highlights their potential as preventative treatments against moulds.

The complete inhibitory effect observed for *M. guilliermondii* in this study aligns with previous reports by Wang et al. [42] and Sepúlveda et al. [44], who noted similar efficacy against two *B. cinerea* strains in vitro. Similarly, the effectiveness of *P. kluyveri* corroborates the findings of Mewa Ngongang et al. [20,45], who also reported its strong inhibitory activity against *B. cinerea*. Additionally, *S. pyralidae* (Y63) demonstrated 100% inhibition against *B. cinerea* B05.10 and IWBT-FF1, and 87% inhibition against *B. cinerea* PPRI 30807. These results are consistent with previous studies by Carbó et al. [46], Ngongang et al. [20], and Gao et al. [47], which showed that various *Candida* spp. exhibit differing degrees of inhibitory activity against *B. cinerea* spoilage.

### 3.4. VOC Production Assay

The mode of action of VOCs in inhibiting *B. cinerea* was explored using the mouth-to-mouth assay. The results indicate that the VOCs produced by the yeasts inhibited the growth of *B. cinerea*, with inhibition levels varying among yeast strains (Figure 7). Notably, the *B. cinerea* strain PPRI 30807 exhibited higher susceptibility (mean inhibition of 73%) to yeast VOCs than strain B05.10 (mean inhibition of 38%). This pattern differed from the dual-culture assay, where the inhibition of PPRI 30807 was less pronounced, suggesting that VOCs could be a primary mode of action against this particular strain.

In the VOC assay trial, *P. kluyveri* (Y64) demonstrated the highest inhibition, achieving 60%, 76%, and 100% inhibition against *B. cinerea* B05.10, IWBT-FF1, and PPRI 30807, respectively (Figure 7). Unlike in the diffusible metabolite assay (Figure 5) where *P. kluyveri* (Y64) exhibited lower inhibition, its stronger performance in the VOC assay points toward VOC production as its primary antagonistic mechanism. This observation aligns with findings by Nägeli et al. [48] who reported *P*. *kluyveri* as an effective inhibitor of *B. cinerea* growth under in vitro conditions, emphasising its reliance on VOCs for mould suppression.

Other yeasts also demonstrated notable VOC-based inhibition. *Suhomyces pyralidae* (Y63) achieved inhibition rates of 32%, 55%, and 82% against strains B05.10, IWBT-FF1, and PPRI 30807, respectively, while *M. guilliermondii* Y88 exhibited 29%, 47%, and 79% inhibition (Figure 7). Previous studies by Mewa-Ngongang et al. [20] and Choińska et al. [49] also reported the effective VOC-mediated inhibition of *B. cinerea* by *S. pyralidae* and *M. guilliermondii,* supporting the current findings. *Zygoascus hellenicus* (Y89) showed moderate inhibition, with rates of 44%, 38%, and 58% against the three *B. cinerea* strains, respectively (Figure 7), further illustrating the potential of VOC production as a biological control strategy against mould proliferation.

### 3.5. VOC Extraction and Gas Chromatographic Analyses

The VOCs produced by *P. kluyveri* and *S. pyralidae* were shown to play a significant role in inhibiting *B. cinerea* growth during in vitro trials (Figure 7). The VOCs produced by *P. kluyveri* and *S. pyralidae* were analysed using solid-phase microextraction coupled with gas chromatography–mass spectrometry (SPME-GC–MS), identifying a total of 29 compounds, of which seven were consistently present across all replicates. The key compounds included alcohols (isobutanol, isoamyl alcohol, 2-phenylethanol), esters (isoamyl acetate, 2-phenethyl acetate), gamma butyrolactone (γ-decanolactone), and a fatty acid methyl ester (methyl palmitate) (Table 4). These VOCs were produced by the yeast isolates alone or in conjunction with *B. cinerea*. Notably, isoamyl alcohol, 2-phenylethanol, 2-phenethyl acetate, and methyl palmitate were also detected when *B. cinerea* was cultured independently.

*Suhomyces pyralidae* produced all seven VOCs in monoculture, with isobutanol, isoamyl acetate and 2-phenethyl acetate concentrations slightly elevated in the presence of *B. cinerea.* These VOCs, particularly isobutanol, isoamyl acetate, and 2-phenethyl acetate, are likely contributors to the inhibition of *B. cinerea*, aligning with findings of Li et al. [50] who reported isoamyl acetate’s antagonistic effects against grey mould on blueberries. Further, Zou et al. [51] demonstrated the antifungal efficacy of isoamyl acetate against *B. cinerea* mycelial growth, and *Hanseniaspora uvarum* effectively controlled *B. cinerea* in strawberries and cherries, with 2-phenylethyl acetate identified as the predominant VOC [52]. Phenylethyl acetate has also shown strong inhibitory activity against *Aspergillus ochraceus* and *Mucor* spp. growth [49,53].

When co-cultured with *B. cinerea*, *S. pyralidae* produced slightly lower levels of isoamyl alcohol and 2-phenylethanol, yet these VOCs continued to contribute to its antagonistic effect. Calvo et al. [54] reported complete inhibition of *B. cinerea* growth in vivo by isoamyl alcohol, with additional studies by Maluleke et al. [31] and Zou et al. [51], linking isoamyl alcohol and 2-phenylethanol to *B. cinerea* inhibition. The observed inhibition of *B. cinerea* may be due to the combined or synergistic effects of VOCs produced by *S. pyralidae*.

*Pichia kluyveri*, when cultured alone, produced high concentrations of isoamyl acetate and 2-phenethyl acetate; however, these levels significantly decreased when co-cultured with *B. cinerea.* Lower concentrations of isoamyl alcohol and 2-phenylethanol were also observed in co-culture, yet these VOCs may still be central to *B. cinerea* suppression.

Previous research on biocontrol yeasts, such as *P. kudriavzevii, P. occidentalis, W. anomalus, H. uvarum*, and *C. intermedia*, demonstrates that VOCs effectively inhibited *B. cinerea* spore germination and mycelial growth [14,31,49,55]. Ethanol and 2-phenylethanol are specifically highlighted as potent antifungal agents against *B. cinerea* and *Alternaria alternata* [56,57]. Additionally, transcinnamaldehyde has shown to inhibitory effects on *B. cinerea* mycelial growth and conidia germination, significantly reducing infections in cherry tomatoes [58]. VOCs such as 3-methyl-1-butanol, 2-phenylethanol, 2-ethyl-1-hexanol, 4-methyl-ethyl ester, and ethyl acetate have also been identified as effective in inhibiting *B. cinerea* spore germination and mycelial growth [59,60,61,62].

The observed decrease in VOC concentrations in the presence of *B. cinerea* may result from interspecies competition for oxygen and/or carbon dioxide within the test environment, with the biocontrol efficacy of the yeast potentially deriving from the synergistic effects of VOCs and elevated carbon dioxide levels [14,63]. Additionally, *B. cinerea* may produce defensive compounds that alter the metabolic pathways of biocontrol yeasts, resulting in shifts in VOC production profiles [64].

### 3.6. Post-Harvest Fruit Bioassays

The application of yeast-based biocontrol agents demonstrated significant efficacy in reducing the spoilage of *B. cinerea* in apples and strawberries, resulting in marked reductions in fruit decay (Figure 8 and Figure 9). In apple trials, *S. pyralidae* (Y63) effectively inhibited the growth of *B. cinerea* strains B05.10, IWBT-FF1, and PPRI 30807 by 64%, 40%, and 25%, respectively (Figure 8). *Aureobasidium melanogenum* displayed inhibitory activity against these strains by 21%, 24%, and 26%, respectively, while *P. kluyveri* achieved inhibition rates of 11%, 17%, and 16%, respectively. In comparison, the commercial fungicide Captan exhibited stronger inhibitory effects, reducing the growth of *B. cinerea* B05.10, IWBT-FF1, and PPRI 30807 by 92%, 59%, and 17%, respectively. These findings align with prior studies. Guerrero Prieto et al. [65] and Carbó et al. [46] reported the antagonistic properties of *Candida* spp., such as *C. oleophila* and *C. sake*, against *B. cinerea* in various fruit contexts, consistent with the inhibitory effects observed for *S. pyralidae* in this study. The activity of *A. melanogenum* corroborates the findings of Di Francesco et al. [33], who demonstrated similar antagonistic effects of *Aureobasidium* species in vivo. Furthermore, *P. kluyveri*’s inhibitory effects and the minimum concentration of 1 × 10^2^ cells/mL required for inhibition by *P. kudriavzevii* against *B. cinerea* are consistent with Maluleke et al. [31], who highlighted the antimicrobial properties of *Pichia* spp. targeting *B. cinerea*.

In strawberry trials, yeast strains and Captan demonstrated greater inhibition, particularly against *B. cinerea* strains B05.10 and IWBT-FF1, compared to their performance in apples. The results indicate that the extent of inhibition is not solely dependent on the specific yeast or mould species but is also influenced by the type of fruit substrate. *Suhomyces pyralidae* (Y63) achieved 100%, 65%, and 34% inhibition against *B. cinerea* strains B05.10, IWBT-FF1, and PPRI 30807, respectively. Similarly, *A. melanogenum* (Y6) exhibited 98%, 95%, and 19% antagonistic activity against the same strains. *Pichia kluyveri* demonstrated 98%, 100%, and 10% inhibition of *B. cinerea* strains B05.10, IWBT-FF1, and PPRI 30807, respectively. Captan, the commercial fungicide, exhibited 100%, 100%, and 23% inhibition against these strains. These findings align with previous studies. The strong inhibitory effects of *Aureobasidium* spp., observed in this study, are consistent with the work of Zajc et al. [66,67], who highlighted the antagonistic capabilities of *Aureobasidium* species under in vivo conditions. The efficacy of *P. kluyveri* supports the findings of Nägeli et al. [48] and Maluleke et al. [31], who demonstrated the potential of *Pichia* spp. in suppressing *B. cinerea*. Furthermore, the high efficacy of Captan observed in this study is consistent with reports by Guerrero Prieto et al. [65], who documented its effectiveness against *B. cinerea* in apples.

Overall, the study indicates that the degree of inhibition is influenced by the specific yeast or mould strain as well as the fruit type. The susceptibility of *B. cinerea* strains varied based on the host fruit, highlighting the importance of fruit type in modulating the antagonistic effects of yeast biocontrol agents. The potential application of these yeast species, either individually or in combination, offers promising alternatives to traditional chemical fungicides for managing *B. cinerea* spoilage, presenting a sustainable solution for the agricultural industry.

## 4. Conclusions

The study confirmed that *A. melanogenum* synthesises enzymes capable of hydrolysing cell walls. In the direct-contact inhibition assays, cell suspensions of *S. pyralidae* and *A. melanogenum* exhibited the strongest antagonistic effects against *B. cinerea,* with the yeasts significantly impeding spore germination. In fruit trials, *S. pyralidae*, *A. melanogenum*, and *P. kluyveri* each demonstrated distinct inhibitory capacities against *B. cinerea*, varying by fruit type, with results comparable to those of commercial fungicides. This suggests that these yeasts hold promise as biocontrol agents for reducing post-harvest spoilage in place of chemical fungicides. The antagonistic mechanism of *P. kluyveri* was linked to the production of VOCs, with isobutanol, isoamyl acetate, and 2-phenethyl acetate identified as key antifungal agents. The primary VOCs contributing to this inhibition were identified as belonging to the alcohol and ester groups. Future investigations should focus on assessing the individual and synergistic antimicrobial effects of these VOCs against *B. cinerea*. Additionally, research should explore the use of biocontrol yeasts in pre-harvest treatments to inhibit mould growth, exploring various yeast combinations and integrating yeast applications with commercial fungicides to mitigate resistance development.

## Figures and Tables

**Figure 1 jof-11-00026-f001:**
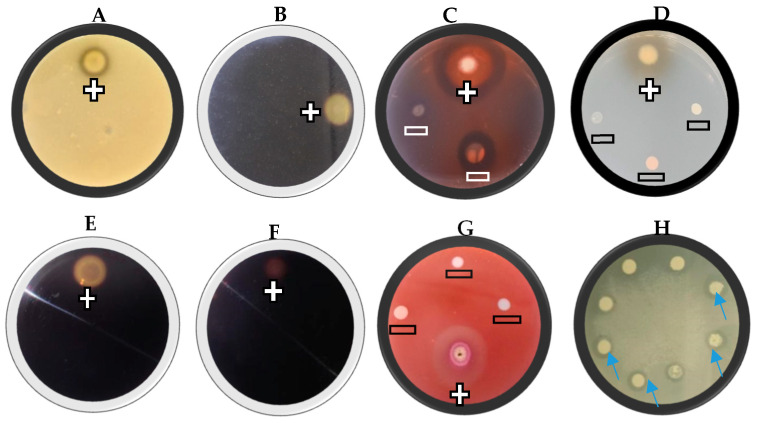
A representative example of the extracellular lytic enzyme activity of selected yeast isolates, proteases (**A**), chitinase (**B**), *β*-1,3-Glucanase (**C**), *β*-glucosidase (**D**), cellulase (**E**), starch (**F**), pectinase (**G**), and lipase (**H**). The positive sign (+) represents enzyme activity and the negative sign (−) represents no enzyme activity. For lipase activity, the arrows show clear halos around the colonies. This is a representative example of three replicates.

**Figure 2 jof-11-00026-f002:**
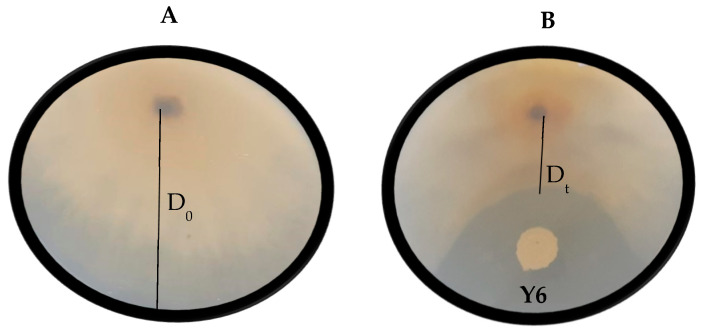
Visual representation of the growth of *Botrytis cinerea* (**A**) and the antagonistic effect of yeast isolate *Aureobasidium melanogenum* (Y6) against *B. cinerea* (**B**) on yeast malt agar. D_0_ represents the horizontal growth of the mould colony on the negative control plates and D_t_ represents the horizontal growth of the mould colony on the yeast-treated plates. Each plate is a representative example of three replicates.

**Figure 3 jof-11-00026-f003:**
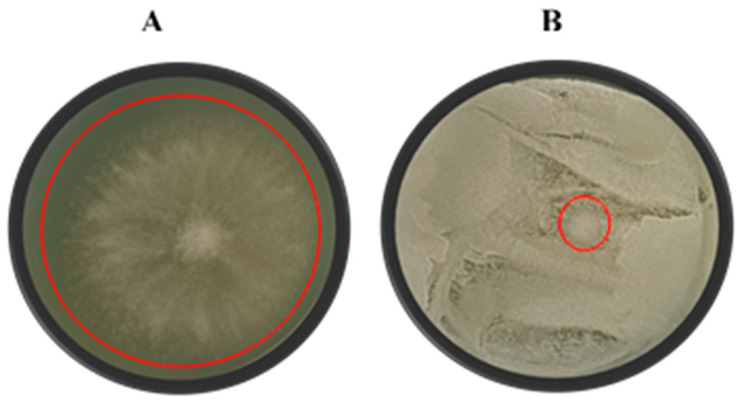
A visual representation of the growth of *Botrytis cinerea* (**A**) and the antagonistic effect of yeast isolate *Suhomyces pyralidae* (Y63) against *B. cinerea* (**B**) on yeast malt agar. D_0_ represents the diameter of the mould colony on the negative control plates and D_t_ represents the diameter of the mould colony on the yeast-treated plates. Each plate is a representative example of three replicates. Red circles show the *B. cinerea* growth for the different treatments.

**Figure 4 jof-11-00026-f004:**
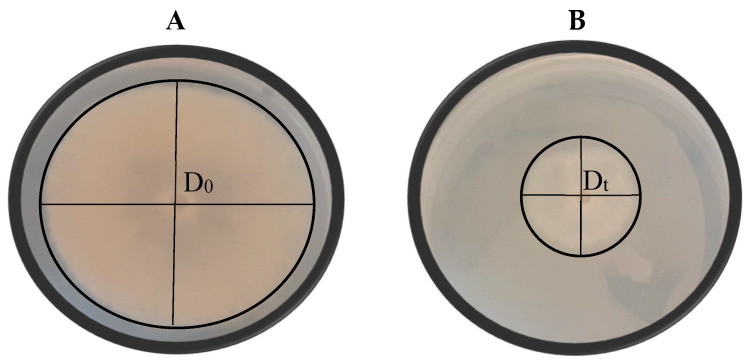
Visual representation of the growth of *Botrytis cinerea* (**A**) and the antagonistic effect of selected yeast isolate *Pichia kluyveri* (Y64) against *B. cinerea* (**B**) on yeast malt agar. D_0_ represents the average diameter of the mould colony on the negative control plates and D_t_ represents the diameter of the mould colony on the treated plates. Each plate is a representative example of three replicates.

**Figure 5 jof-11-00026-f005:**
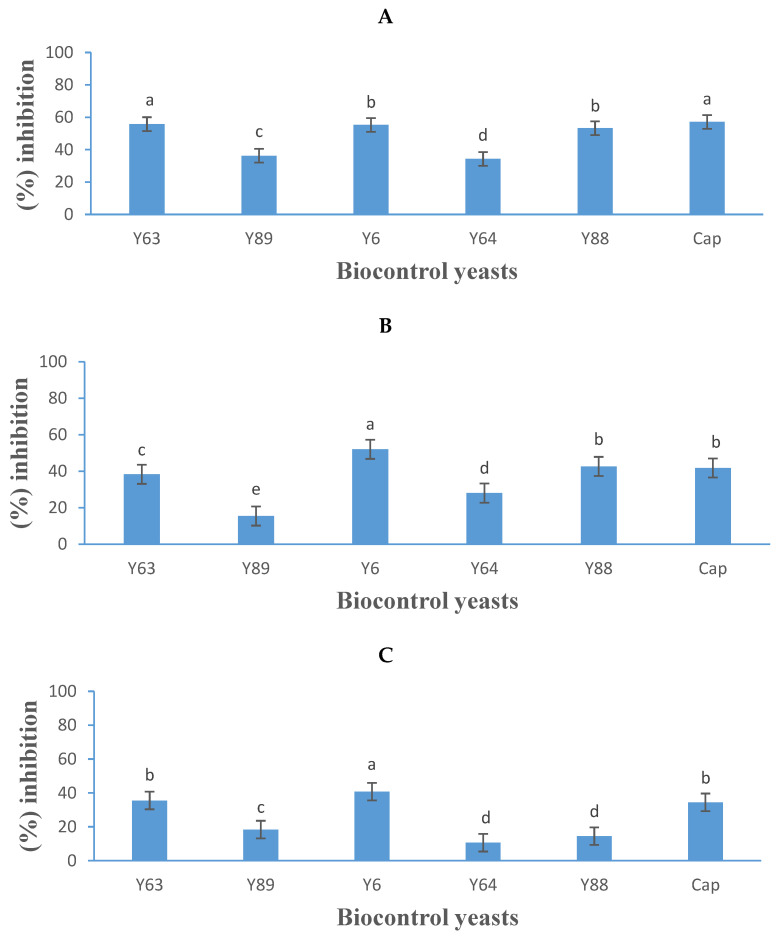
Growth inhibition activity (expressed as a percentage) of five yeasts (details listed in Table 2) and Captan (Cap), a commercial fungicide against *Botrytis cinerea* B05.10 (**A**), IWBT-FF1 (**B**), and PPRI 30807 (**C**) using the dual-culture assay. Values are means of three replicates and the standard deviations are also shown. Different letters indicate significant differences (*p ≤* 0.05) between treatments. The plates of negative control treatments only contained the respective moulds and were the references for determining growth inhibition.

**Figure 6 jof-11-00026-f006:**
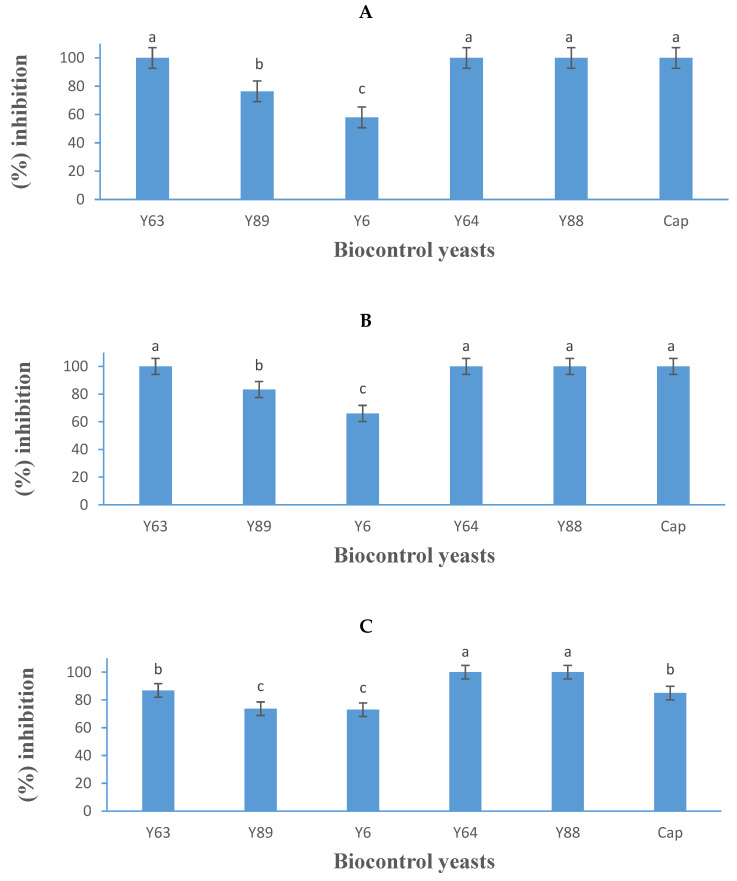
Growth inhibition activity (expressed as a percentage) of five yeasts (details listed in Table 2) and Captan (Cap), a commercial fungicide against *Botrytis cinerea* B05.10 (**A**), IWBT-FF1 (**B**), and PPRI 30807 (**C**), using the mould spore germination assay. Values are means of three replicates and the standard deviations are also shown. Different letters indicate significant differences (*p* ≤ 0.05) between treatments. The plates of negative control treatments only contained the respective moulds and were the references for determining growth inhibition.

**Figure 7 jof-11-00026-f007:**
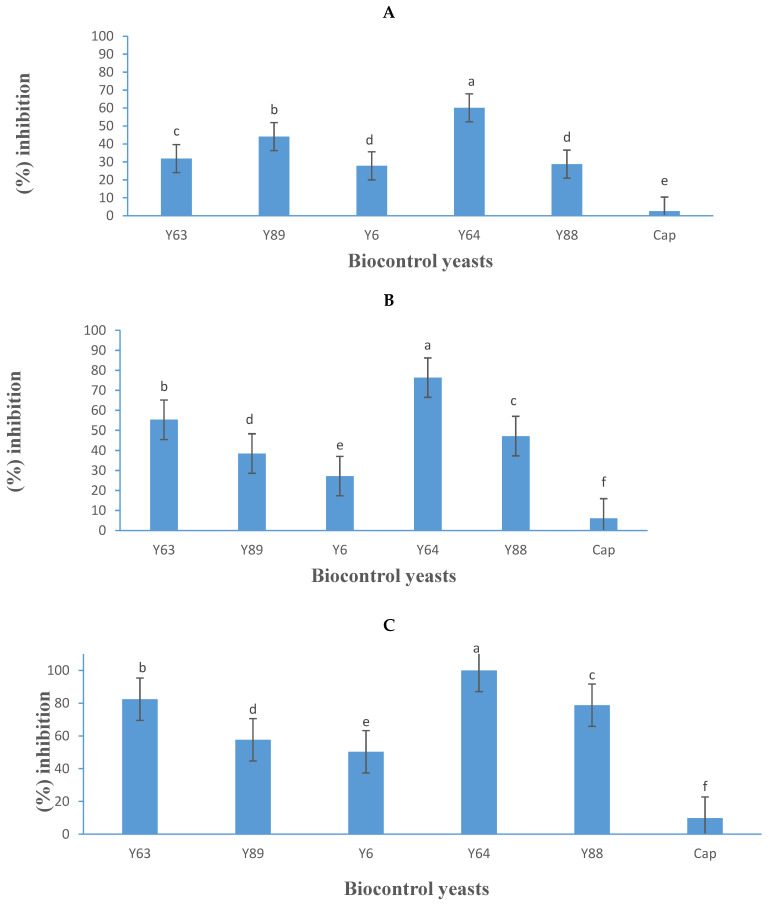
Growth inhibition activity (expressed as a percentage) of five yeasts (details listed in Table 2) and Captan (Cap), a commercial fungicide against *Botrytis cinerea* B05.10 (**A**), IWBT-FF1 (**B**), and PPRI 30807 (**C**) based on volatile organic compound production. Values are means of three replicates and the standard deviations are also shown. Different letters indicate significant differences (*p* ≤ 0.05). The negative control treatments only contained the respective moulds and were the references for determining growth inhibition.

**Figure 8 jof-11-00026-f008:**
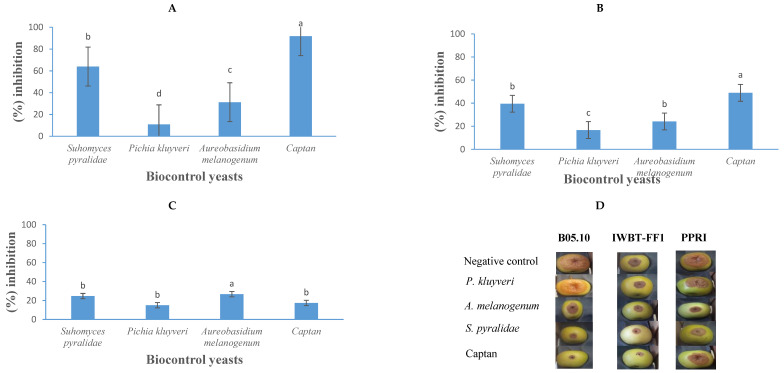
The growth inhibition activity (%) of *Suhomyces pyralidae* (Y63)*, Aureobasidium melanogenum* (Y6), and *Pichia kluyveri* (Y64) against *Botrytis cinerea* B05.10 (**A**), IWBT-FF1 (**B**), and PPRI 30807 (**C**) during post-harvest trials on apples. Values are means of five replicates and the standard deviations are also shown. The different letters indicate significant differences (*p* ≤ 0.05) between treatments. (**D**) Photographs of apples showing lesion diameters. Each set is a representative example of 25 apples. For the negative control treatments, the apples were only infected with the respective moulds, therefore no growth inhibition was observed.

**Figure 9 jof-11-00026-f009:**
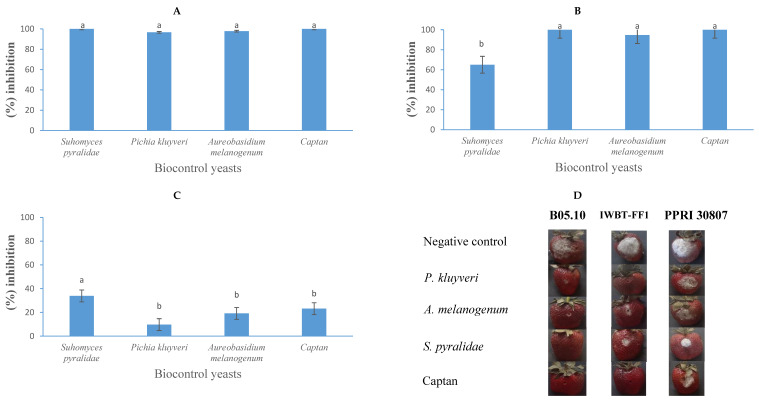
The growth inhibition activity (expressed as a percentage) of *Suhomyces pyralidae* (Y63)*, Aureobasidium melanogenum* (Y6), and *Pichia kluyveri* (Y64) against *Botrytis cinerea* B05.10 (**A**), IWBT-FF1 (**B**), and PPRI 30807 (**C**) during post-harvest trials on strawberries. Values are means of five replicates and the standard deviations are also shown. The different letters indicate significant differences (*p* < 0.05) between treatments. (**D**) Photographs of strawberries showing lesion diameters. Each set is a representative example of 25 strawberries. For the negative control treatments, the strawberries were only infected with the respective moulds, therefore no growth inhibition was observed.

**Table 1 jof-11-00026-t001:** Yeasts screened for the production of lytic enzymes.

Yeast Code	Species Name	Starch	Cellulase	Protease	Glucosidase	Chitinase	Pectinase	Lipase	*β*-1,3 Glucanase
Y6	*Aureobasidium melanogenum*	+ *	+	+	+	+	+	+	+
Y11	*Debaryomyces hansenii*	−	−	−	−	−	−	+	−
Y17	*Hanseniaspora occidentalis*	−	−	−	−	−	−	+	−
Y24	*Meyerozyma guilliermondii*	−	−	−	−	−	−	+	−
Y35	*Rhodotorula dairenensis*	−	−	+	−	−	−	+	+
Y39	*M. guilliermondii*	−	−	−	−	−	−	+	−
Y63	*Suhomyces pyralidae*	+	+	+	−	−	−	+	+
Y64	*Pichia kluyveri*	−	−	−	−	−	−	+	−
Y65	*M. guilliermondii*	−	−	+	−	−	−	+	+
Y74	*Torulaspora delbrueckii*	−	−	−	−	−	−	+	+
Y75	*Saccharomyces cerevisiae*	−	−	−	−	−	−	+	−
Y83	*Brettanomyces bruxellensis*	−	−	−	−	−	−	+	−
Y84	*D. hansenii*	−	−	−	−	−	−	+	−
Y88	*M. guilliermondii*	+	+	+	−	−	−	+	+
Y89	*Zygoascus hellenicus*	+	+	+	−	−	−	+	+
Y91	*Zygosaccharomyces rouxii*	−	−	−	−	−	−	+	−
Y92	*Z. rouxii*	−	−	−	−	−	−	+	−
Y93	*Z. microellipsoides*	−	−	−	−	−	−	+	−
Y95	*Z. florentinus*	−	−	−	−	−	−	+	−
Y96	*Z. fermentati*	−	−	−	−	−	−	+	−
Y97	*Z. bisporus*	−	−	−	−	−	−	+	−
Y102	*Starmerella magnoliae*	−	−	−	−	−	−	+	−
Y103	*Saccharromyces cerevisiae*	−	−	−	−	−	−	+	−

* (−) no enzyme activity, (+) enzyme activity.

**Table 2 jof-11-00026-t002:** Yeasts selected for the dual assays, mould spore germination, and mouth-to-mouth assays on yeast malt agar.

Yeast Code	Species Name	Origin
Y6	*Aureobasidium melanogenum*	Jaboticaba fruit
Y63	*Suhomyces pyralidae*	Shiraz wine fermentation
Y64	*Pichia kluyveri*	Shiraz wine fermentation
Y88	*Meyerozyma guilliermondii*	Apple
Y89	*Zygoascus hellenicus*	Apple

**Table 3 jof-11-00026-t003:** Treatments applied on apples and strawberries during post-harvest biocontrol trials.

Treatment	Description
Treatment 1	Sterile distilled water (Control)
Treatment 2	*Botrytis cinerea* B05.10
Treatment 3	*B. cinerea* IWBT-FF1
Treatment 4	*B. cinerea* PPRI 30807
Treatment 5	*B*. *cinerea* B05.10 and *Suhomyces pyralidae* Y63
Treatment 6	*B*. *cinerea* IWBT-FF1 and *S. pyralidae* Y63
Treatment 7	*B*. *cinerea* PPRI 30807 and *S. pyralidae* Y63
Treatment 8	*B*. *cinerea* B05.10 and *Pichia kluyveri* Y64
Treatment 9	*B*. *cinerea* IWBT-FF1 and *P. kluyveri* Y64
Treatment 10	*B*. *cinerea* PPRI 30807 and *P. kluyveri* Y64
Treatment 11	*B*. *cinerea* B05.10 and *Aureobasidium melanogenum* Y6
Treatment 12	*B*. *cinerea* IWBT-FF1 and *A. melanogenum* Y6
Treatment 13	*B*. *cinerea* PPRI 30807 and *A. melanogenum* Y6
Treatment 14	*B*. *cinerea* B05.10 and Captan
Treatment 15	*B*. *cinerea* IWBT-FF1 and Captan
Treatment 16	*B*. *cinerea* PPRI 30807 and Captan

**Table 4 jof-11-00026-t004:** Major volatile compounds (VOCs) produced by *Pichia kluyveri*, *Suhomyces pyralidae*, and *Botrytis cinerea* PPRI 30807, identified through gas chromatography–mass spectrometry (GC-MS).

VOCs	*B. cinerea*	*P. kluyveri*	*P. kluyveri* and *B. cinerea*	*S. pyralidae*	*S. pyralidae* and *B. cinerea*
	Average Area Ratio				
Isobutanol	ND *	0.006	ND	0.009	0.012
Isoamyl acetate	ND	2.489	ND	0.001	0.002
Isoamyl alcohol	0.006	0.136	0.040	0.140	0.135
2-Phenethyl acetate	0.004	3.207	1.325	0.002	0.004
2-Phenylethanol	0.005	0.155	0.034	0.021	0.019
γ-Decanolactone	ND	ND	ND	0.008	0.003
Methyl palmitate	0.004	0.012	0.001	0.001	0.001

* ND-Not detected.

## Data Availability

The original contributions presented in the study are included in the article, further inquiries can be directed to the corresponding authors.

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
