# Peer review of "The Use of Specific Non-Saccharomyces Yeasts as Sustainable Biocontrol Solutions Against Botrytis cinerea on Apples and Strawberries"

_jof, 2025, doi:10.3390/jof11010026_

Round 1
Reviewer 1 Report
Manuscript entitled ‘’Non-Saccharomyces yeasts as sustainable biocontrol solutions against Botrytis cinerea in apples and strawberries”. The manuscript selected three yeast species as biological control agents against B. cinerea. Minor points need to be addressed.
1. I suggested change the current title to “’Screening yeasts as sustainable biocontrol solutions against Botrytis cinerea in apples and strawberries”.
2. Line 3. Italic for Botrytis cinerea.
3. Line 277-279. Did the author use disease index and biocontrol efficiency to present post-harvest fruit bioassays?
Manuscript entitled ‘’Non-Saccharomyces yeasts as sustainable biocontrol solutions against Botrytis cinerea in apples and strawberries”. The manuscript selected three yeast species as biological control agents against B. cinerea. Minor points need to be addressed.
1. I suggested change the current title to “’Screening yeasts as sustainable biocontrol solutions against Botrytis cinerea in apples and strawberries”.
2. Line 3. Italic for Botrytis cinerea.
3. Line 277-279. Did the author use disease index and biocontrol efficiency to present post-harvest fruit bioassays?
Reviewer 2 Report
The relevance of the work and the results obtained by the authors is beyond doubt, first of all, from a practical point of view, and the authors demonstrate this quite convincingly in the introductory part of their work. There is also no doubt that this work and its results are of great interest from a scientific point of view, as they reveal some new aspects of the biotic relationships of fungi, in particular, the role of gaseous compounds in them.
The relevance of the work and the results obtained by the authors is beyond doubt, first of all, from a practical point of view, and the authors demonstrate this quite convincingly in the introductory part of their work. There is also no doubt that this work and its results are of great interest from a scientific point of view, as they reveal some new aspects of the biotic relationships of fungi, in particular, the role of gaseous compounds in them.
The analysis of the manuscript shows that the abstract gives a fairly complete picture of the purpose and results of the work. In the introduction, the authors focus on the practical aspects of the relevance of the study, ignoring the issues of its high scientific significance. The Materials and Methods section is well prepared and illustrated. It shows that the authors used a variety of complementary methods of analysis, each of which is briefly but meaningfully described. I have only one question about this section: why did the authors not consider it possible to explain in more detail the choice of objects (strains) of the study? The main questions concern the main section of the work Results and Discussion. Already at the very beginning there is a link to Table 1, but it is in the Materials and Methods section. If its data were obtained by the authors of the work, then its place is in the Results and Discussion section, if not, then in the Materials and Methods section!
Subsection 3.2 discusses the activity of five strains against Botrytis cinerea, but in the text, when discussing the results, only 4 are discussed: Y6, Y63, Y88, Y64. At the same time, the author's own data are discussed very briefly and then literature data are cited. The result is a bizarre mixture of author's and literature data. This is also typical for sections 3.3, 3.6. I think it would be better to first present the author's original data and then briefly discuss them using literature data. The conclusion is prepared briefly, but its provisions logically follow from the materials of the article: three yeast strains most aggressive towards Botrytis cinerea are noted, as well as prospects for further research.
I think that the article can be published, preferably taking into account the reviewer's comments.
The relevance of the work and the results obtained by the authors is beyond doubt, first of all, from a practical point of view, and the authors demonstrate this quite convincingly in the introductory part of their work. There is also no doubt that this work and its results are of great interest from a scientific point of view, as they reveal some new aspects of the biotic relationships of fungi, in particular, the role of gaseous compounds in them.
The analysis of the manuscript shows that the abstract gives a fairly complete picture of the purpose and results of the work. In the introduction, the authors focus on the practical aspects of the relevance of the study, ignoring the issues of its high scientific significance. The Materials and Methods section is well prepared and illustrated. It shows that the authors used a variety of complementary methods of analysis, each of which is briefly but meaningfully described. I have only one question about this section: why did the authors not consider it possible to explain in more detail the choice of objects (strains) of the study? The main questions concern the main section of the work Results and Discussion. Already at the very beginning there is a link to Table 1, but it is in the Materials and Methods section. If its data were obtained by the authors of the work, then its place is in the Results and Discussion section, if not, then in the Materials and Methods section!
Subsection 3.2 discusses the activity of five strains against Botrytis cinerea, but in the text, when discussing the results, only 4 are discussed: Y6, Y63, Y88, Y64. At the same time, the author's own data are discussed very briefly and then literature data are cited. The result is a bizarre mixture of author's and literature data. This is also typical for sections 3.3, 3.6. I think it would be better to first present the author's original data and then briefly discuss them using literature data. The conclusion is prepared briefly, but its provisions logically follow from the materials of the article: three yeast strains most aggressive towards Botrytis cinerea are noted, as well as prospects for further research.
I think that the article can be published, preferably taking into account the reviewer's comments.

Reviewer 3 Report
The study offers valuable insights into the biological control of gray mold in strawberries and apples using yeasts.
Although the manuscript is well-written, there are a few areas that require improvement:
- Some scientific names of fungi are outdated.
- Certain names are not written in italics, and some are not abbreviated after the first citation.
- Some aspects of the methodology need to be further detailed.
- Figure 4 cannot be assessed, as it was not included in the review file.
